# Spirituality and Cyberaggression: Mediating and Moderating Effect of Self-Control and School Climate

**DOI:** 10.3390/ijerph20042973

**Published:** 2023-02-08

**Authors:** Shengnan Li, Yangang Nie

**Affiliations:** Department of Psychology and Research Center of Adolescent Psychology and Behavior, School of Education, Guangzhou University, Guangzhou 510006, China

**Keywords:** spirituality, self-control, cyberaggression, school climate, students

## Abstract

Background: Cyberaggression is an essential topic to focus on when it comes to adolescents’ development. We focused on understanding the relationship between spirituality, self-control, school climate, and cyberaggression by examining the mediating and moderating effect of self-control and school climate. Methods: We examined 456 middle school students (M age = 13.45, SD = 1.07), 475 high school students (M age = 16.35, SD = 0.76), and 1117 college students (M age = 20.22, SD = 1.50). Results: Results indicated that the mediating effect of self-control was significant for the college sample on both types of cyberaggression and marginally significant for the high school and middle school sample on reactive cyberaggression. The moderating effect varied across the three samples. School climate moderated the first half of the mediation model for all three samples, the second half for middle school and college student samples on reactive cyberaggression, the direct path for middle school samples on reactive cyberaggression, and the college student sample on both types of cyberaggression. Conclusion: Spirituality has varying degrees of association with cyberaggression through the mediating role of self-control and the moderating role of school climate.

## 1. Introduction

Cyberaggression has been studied quite extensively since the internet has become ubiquitous in our everyday life during the past few decades. The negative impact of cyberaggression includes a harmful impact on adolescents’ psychological health, outlook on values, and moral development [1,2]. With this reality, it is imperative that researchers and practitioners understand important factors that could impact cyberaggression and effective interventions to manage it. Although cognitive resources such as self-control have been identified as important factors that individuals could utilize to manage aggression [3,4], the results of many studies are confined within a laboratory setting and have limited transferability to the real world [5].

Spirituality has been found to promote self-control [6]. Spirituality among Chinese college students is a concept that encompasses one’s value system and could provide a source of psychological power [7,8]. However, this important concept has yet to be considered by researchers to potentially manage aggression. In addition, although school climate is a protective factor for many aspects of adolescents’ development, including decreasing cyberaggression [9,10], whether school climate further enhances the positive influence of spirituality on cyberaggression through self-control is unknown. Our current study focuses on understanding the relationship between spirituality, self-control, cyberaggression, and school climate in order to navigate another potentially helpful way to curb cyberaggression. 

## 2. Theoretical Background 

### 2.1. Cyberaggression and Self-Control

For traditional aggressive behaviors, [11] postulated the general aggression model (GAM) after integrating previous researchers’ theories. This theory states that aggressive behaviors consist of three factors: cognition, affect, and arousal. These three factors are interconnected and guide each other. For instance, aggressive cognition can increase aggressive affect and vice versa.

Cyber aggressive behaviors are behaviors that can cause intentional harm to individuals’ reputations using computer-mediated communication [12,13]. Similar to traditional aggressive behaviors, cyber aggressive behaviors are also offensive, derogatory, harmful, and unwanted [2]. Although cyberbullying is often used in the literature to describe above mentioned behaviors, cyberaggression is considered a more appropriate term than cyberbullying [14] because specific factors, such as inherent power imbalance, are not always present to qualify the definition of bullying within cyberaggression [15,16,17]. Although some believe that types of cyberaggression include picture-based/physical, verbal/text-based, and relational/texted-based [14], a more popular differentiation of cyberaggression is to include instrumental and reactive aspects [18,19,20]. Specifically, the instrumental aspect helps individuals deliver cyberaggression in a more active form and relates to the overt form of cyberaggression, such as the direct attack on one’s personality, while the reactive aspect often occurs when individuals react and tackle the relational part, such as that cyberaggression was a means to fulfill the purpose of relational retaliation. 

Interventions to address cyberaggression often utilize cognitive strategies. One such cognitive resource to consider in order to reduce cyberaggression is self-control, which is the ability to “override or change one’s inner responses, as well as to interrupt undesirable behavioral tendencies (such as impulses) and refrain from acting on them” ([21], p. 274). Self-control is affected by many factors, including individual and environmental factors. Individual factors include dispositions, such as personality and characteristics, while environmental factors include family, school, peers, neighbors, and cultural atmosphere [22]. It has been shown that individuals who have high levels of self-control tend to demonstrate more pro-social behaviors [23], and self-control was found to be negatively related to cyberbullying [24] and cyberaggression [25]. 

### 2.2. The Motivating Effect of Spirituality on Self-Control and Self-Control’s Mediating Role

Since self-control is helpful in curbing the level of aggression, such as cyberaggression, it is natural to consider promoting individuals’ level of self-control with the purpose of reducing cyberaggression. Spirituality is a Western concept that indicates one’s connection with a higher power. One’s chosen values and practice. This connection can bring one a sense of meaning in life and help one become more loving [26,27]. Consequently, it would lead individuals to conduct more prosocial behaviors and less aggressive ones. However, it is very rare in the literature that researchers have studied such a relationship. The only two studies that we encountered were regarding religiosity and self-control. 

In one study, researchers used the word scramble paradigm in experimental studies of religiosity and self-control [28]. Specifically, concepts that are related to religiosity and neutral concepts are scrambled and are available for participants to choose from and compose some meaningful sentences. Participants demonstrated higher levels of self-control after being primed with “God” related concepts than if they were primed with neutral concepts and conducted more prosocial behaviors [29]. In the other study, researchers found that activating participants’ religiosity can promote self-control in the endurance of discomfort, delay of gratification, persistence at a task, and cognitive control [6]. Based on the close relationship between spirituality and religiosity on the conceptual level [30], we believe that the positive relationship between spirituality and self-control could potentially exist as well. 

### 2.3. Spirituality in Chinese Context

Prior research has attempted to examine the spirituality of Chinese college students. Ref. [7] found that Chinese college students were not familiar with the literal translation of spirituality; however, when someone mentioned Xin-Yang, a Chinese concept concerning one’s values and beliefs, participants were more comfortable and started talking more. Ref. [8] surveyed over 2000 college students and developed the Xin-Yang Scale for Chinese college students that are both valid and reliable. Ref. [8] found that college students’ spirituality does not point toward any specific religious or political beliefs, and its target could be any object that could bring mental guidance. The scale is composed of three dimensions: Mental Guidance, Relationship to Others, and Characteristics. These three dimensions essentially have conceptual overlapping with that of Western spirituality.

The three dimensions of Chinese college students’ spirituality include mental guidance, characteristics, and relationship with others. Chinese college students’ spirituality can provide people with spiritual motivation and meaning in life. People who have higher levels of spirituality are consistent with higher levels of kindness, trustworthiness, responsibility, and devotion; they care more about their relationship with others, and they hold high levels of moral standards toward themselves [7]. In order to avoid confusion, spirituality is used in the current study even though Xin-Yang might be the more appropriate label for Chinese spirituality. Therefore, we postulate that a high level of spirituality is associated with more prosocial behaviors, hence a lower level of cyberaggression. No other published studies have been found that examined the relationship between spirituality and self-control and self-control’s role in reducing aggression. We further postulate that spirituality can motivate self-control when one’s level of self-control is low and consequently help one reduce cyberaggression. 

With the information from the previous section that indicated a negative correlation between self-control and cyberaggression, we postulate that there is a potential mediating relationship between spirituality, self-control, and cyberaggression. Specifically, the curbing effect that spirituality has on cyberaggression is mediated through self-control.

### 2.4. School Climate’ Moderating Effect

School is an important environment for adolescents’ development and growth. School climate is defined as students perceived interpersonal relationship quality and opportunity to develop autonomy [31]. Studies have confirmed the protective effect of school climate on many aspects of adolescents’ development [32,33]. Ref. [34] found that school climate (including empathy, teachers’ support, and peer interaction) positively predicted individuals’ defending behaviors when encountering cyberbullying. In addition, a poor school climate is associated with increased bullying, and it is very important to consider the school climate when schools try to intervene in cases of cyberaggression [35,36]. It indicates that different aspects of school climate affect cyberaggression differently. 

There is no research that directly provided evidence for the moderating effect of school climate on the relationship between self-control and cyberaggression. However, researchers have claimed that it is possible a positively perceived school climate could help mitigate subsequent aggression; and because of this, individuals’ self-control resource is deployed to help them be on task and motivated to learn instead of being used to curve aggression [37].

Similarly, no published study examined school climate as a moderator for the relationship between spirituality, self-control, and cyberaggression. However, existing research has provided a solid foundation for the possibility. Based on Bronfenbrenner’s Ecological System Theory, Ref. [38] proposed the System View of School Climate theory, in which a nanosystem was introduced. A nanosystem includes units such as peers and classes that are embedded in schools, and it interacts with other nanosystems as well as microsystems. Within this nanosystem, individual students interact with their peers and class, and in turn, their peers and class exert influence on various aspects, including their value systems, such as spirituality. Consequently, a microsystem such as a school climate could indirectly affect these individual students’ value systems through nanosystems. Therefore, we propose that this important environmental variable and microsystem—school climate can affect spirituality directly and indirectly through its relationship with the nanosystems and consequently affects cyberaggression.

### 2.5. Current Study

Spirituality is a valuable psychological resource and outlook on values. Although it has a conceptual similarity with that of religiosity, whether spirituality has a similar motivating effect to self-control is currently unknown. What is more, school climate is an important protective factor in adolescents’ developmental process. Nevertheless, how this environmental variable function when it comes to the relationship between individual variables such as spirituality, self-control, and cyberaggression remains unclear. Therefore, our study has the following hypotheses: (1) spirituality is negatively related to cyberaggression; (2) self-control would mediate the relationship between spirituality and cyberaggression; and within the mediation model, (3) school climate would moderate the relationship between spirituality and cyberaggression (direct path); (4) school climate moderates the relationship between spirituality and self-control (first half of the mediation model), and (5) school climate moderates the relationship between self-control and cyberaggression (second half of the mediation model). See Figure 1 for a demonstration of our hypotheses.

## 3. Methods

### Procedure and Participants

The current study has been approved by the research ethics board at BLINDED FOR REVIEW (Protocol Number GZHU20210001). Prior to data collection, two English study measures were translated from English into Chinese, following the back-translation process [39]. We collected data from two middle schools, two high schools, and two universities located in southern and central China. Participants completed the consent form prior to data collection, which specified the purpose of the study, potential benefits, and risks of participation, and that participation would include the completion of a number of survey items about values and beliefs, their internet use behaviors, their behavioral characteristics, and their school environment. Participants were also informed that participation was voluntary and that they could stop participation at any time during the data collection process. 

A total of 456 middle school students data were valid. Their mean age was 13.45 (SD = 1.07). Of these students, 238 (52.2%) were female, and 218 (47.8%) were male; 247 (54.2%) were in their first year, and 209 (45.8%) were in their second year. In terms of the number of siblings, 230 (50.4%) reported being an only child, 197 (43.2%) reported having one sibling, and 29 (6.4%) reported having two or more siblings. For their parents’ monthly income, 10 (2.2%) earned 1000 CNY or below, 100 (21.9%) earned 1000 to 5000 CNY, 159 (34.9%) earned 5000 to 10,000 CNY, and 187 (41%) earned more than 10,000 CNY. For the parent’s educational level, 7 (1.5%) had education in elementary school or below, 21 (4.6%) had education in middle school or below, 99 (21.7%) had a high school diploma or below, 258 (56.6%) had a Bachelor’s degree or below, 51 (11.2%) had a Master’s degree or below, and 20 (4.4%) had a doctoral degree or below. 

For high school students, a total of 475 students’ data were valid. Their mean age was 16.35 (SD = 0.76). Of these students, 281 (59.2%) were female, and 194 (40.8%) were male; 242 (50.9%) were in their first year, and 233 (49.1%) were in their second year. In terms of the number of siblings, 210 (44.2%) reported being an only child, 214 (45.1%) reported having one sibling, and 51 (10.7%) reported having two or more siblings. For their parents’ monthly income, 13 (2.7%) earned 1000 CNY or below, 131 (27.6%) earned 1000 to 5000 CNY, 177 (37.3%) earned 5000 to 10,000 CNY, and 154 (32.4%) earned more than 10,000 CNY. For the parent’s educational level, 12 (2.5%) had education in elementary school or below, 72 (15.2%) had education in middle school or below, 171 (36%) had a high school diploma or below, 178 (37.5%) had a Bachelor’s degree or below, 29 (6.2%) had a Master’s degree or below, and 13 (2.7%) had a doctoral degree or below. 

We also have 1117 valid cases from the college student sample. Their mean age was 20.22 (SD = 1.50). Of these students, 738 (66.1%) were female, and 379 (33.9%) were male; 425 (38%) were freshmen, 519 (46.5%) were sophomores, 124 (11.1%) were junior, 45 (4%) were senior, and 4 (0.4%) reported to be other. In terms of the number of siblings, 256 (22.9%) reported being an only child, 597 (53.4%) reported having one sibling, and 264 (23.6%) reported having two or more siblings. For their parents’ monthly income, 130 (11.6%) earned 1000 CNY or below, 685 (61.3%) earned 1000 to 5000 CNY, 210 (18.8%) earned 5000 to 10,000 CNY, and 92 (8.2%) earned more than 10,000 CNY. For the parent’s educational level, 193 (17.3%) had an education in elementary school or below, 429 (38.4%) had an education in middle school or below, 334 (29.9%) had a high school diploma or below, 154 (13.8%) had a Bachelor’s degree or below, 2 (0.2%) had a Master’s degree or below, and 5 (0.4%) had a doctoral degree or below. See Table 1 for details. 

## 4. Measures

### 4.1. Chinese College Students Spirituality Scale [8]

This scale is composed of three dimensions: mental guidance, relationship to others, and characteristics. It has 31 items, and items are rated on a 5-point Likert-type scale ranging from (1 = *does not apply* to 5 = *totally apply*). A higher score indicates a higher level of spirituality. Previous research reported acceptable reliability (Cronbach’s alpha is 0.62–0.94, [8]). Cronbach’s alpha for this study is 0.98.

### 4.2. Brief Self-Control Scale [21]

This scale is used to assess adolescents’ level of self-control and has 13 items. Items are rated on a 5-point Likert-type scale ranging from (1 = *not at all like me* to 5 = *very much like me*). A higher score indicates a higher level of self-control. Previous research reported acceptable reliability, and Cronbach’s alpha is 0.83–0.85 [21]. Cronbach’s alpha for this study is 0.84.

### 4.3. Cyber-Aggressive Behaviors Scale [20]

This scale is to measure individuals’ cyber-aggressive behaviors and is composed of two dimensions: instrumental aggression and reactive aggression. It has 31 items, and items are rated on a Likert-type scale ranging from (1 = *never* to 4 = *always*). A higher score indicates a higher level of cyberaggression. Previous research reported acceptable reliability, and Cronbach’s alpha is 0.83–0.86 [20]. Cronbach’s alpha for this study is 0.99.

### 4.4. Perceived School Climate Scale [40]

The school climate measure is a revised 25-item version of two school climate measures, and it assesses three dimensions of school climate: teacher support, student–student support, and opportunities for autonomy in the classroom. All items are rated on a 4-point Likert-type scale ranging from (1 = *never*, 4 = *always*). A higher score indicates higher levels of support or opportunities for autonomy in the classroom. Previous research reported acceptable reliability, and Cronbach’s alpha is 0.81–0.84 [40]. Cronbach’s alpha for this study is 0.92.

### 4.5. Data Analysis 

We removed participants who did not correctly answer the validity check item, for which we asked participants to choose “*totally apply*” for the item. After that, we removed participants who appeared to answer the questions with a pattern, where they answered most questions with the same number. We reverse-coded negatively worded items in the Perceived School Climate Scale and Brief Self-Control Scale. We added a bi-factor for negatively worded items due to low factor loadings of items in the Perceived School Climate Scale. We used SPSS23 Process and Mplus 7.4 to conduct the data analyses, specifically the mediation and moderated mediation analysis. Based on [41] sample size calculation for structural equation modeling, we used [42] calculation tool and determined that the minimum sample size required for our study is 341 to achieve a power of 0.80 and an effect size of 0.40, and our current sample size is 2048 in total.

In the current study, we chose to conduct analyses for three samples separately due to the following reasons. Firstly, the Chinese college spirituality scale was developed among a sample of Chinese college students. In order to gather valid evidence from other age groups, we need to conduct analyses separately. Secondly, the sample size of these three groups is different; combining the samples into a whole might not provide valuable information regarding the relationship between variables. 

## 5. Results

### 5.1. Preliminary Analysis

Correlations between variables are provided in Table 2. Spirituality is positively and significantly correlated with Self-control (*r* = 0.51, *p* < 0.001) and School Climate (*r* = 0.62, *p* < 0.001), and it is negatively and significantly correlated with Instrumental Aggression (*r* = −0.11, *p* < 0.001) and Reactive Aggression subscale (*r* = −0.12, *p* < 0.001). Self-Control is positively and significantly correlated with School Climate (*r* = 0.45, *p* < 0.001) and is negatively and significantly correlated with the Instrumental Aggression (*r* = −0.22, *p* < 0.001) and Reactive Aggression subscale (*r* = −0.22, *p* < 0.001). School Climate is negatively and significantly correlated with the Instrumental Aggression (*r* = −0.20, *p* < 0.001) and Reactive Aggression subscale (*r* = −0.20, *p* < 0.001). Lastly, the Instrumental Aggression and Reactive Aggression subscale are negatively and significantly correlated with each other (*r* = −0.93, *p* < 0.001).

### 5.2. Mediation and Moderation

The direct relation between gender and instrumental cyberaggression is significant for middle school, high school, and college samples (β = 0.14, *p* = 0.001; β = 0.16, *p* < 0.001; β = 0.18, *p* < 0.001), and its relation with reactive cyberaggression is also significant for all three samples (β = 0.14, *p* = 0.003; β = 0.15, *p* < 0.001; β = 0.17, *p* < 0.001). The direct relation between spirituality and instrumental cyberaggression is only significant for the college sample (β = −0.05, *p* < 0.001), and between spirituality and reactive cyberaggression is significant for the college sample (β = −0.05, *p* < 0.001) and almost significant for middle school and high school samples (β = −0.22, *p* = 0.05; β = 0.07, *p* = 0.06). 

Based on the results, we decided to analyze the mediating effect of self-control on both types of cyberaggression for the college student sample and on reaction cyberaggression for middle and high school student samples. We found that self-control significantly mediated all relations after controlling for self-control and all the demographic variables (gender, age, grade, monthly income, parental educational level, and the number of siblings) on instrumental (β = −0.26, *p* = 0.02) for college sample and reactive cyberaggression (β = −0.40, *p* = 0.001; β = −0.54, *p* < 0.001; β = −0.31, *p* = 0.002) for these three samples.

We also conducted moderated mediation analyses for middle school, high school, and college samples separately. School climate moderated the direct path from spirituality to reactive cyberaggression (β = 0.24, *p* < 0.001) for the middle school sample; for the college sample, school climate moderated the direct path from spirituality to both instrumental (β = 0.13, *p* < 0.001) and reactive cyberaggression (β = 0.13, *p* < 0.001); for high school sample, school climate did not moderate the direct path from spirituality to reactive cyberaggression (β = 0.04, *p* = 0.19). For the fourth hypothesis, school climate moderated the relationship between spirituality and self-control (β = 0.16, *p* = 0.001) for the middle school sample, for the high school sample (β = 0.39, *p* < 0.001), and for the college sample (β = 0.32, *p* < 0.001).

For the fifth hypothesis, school climate moderated the relationship between self-control and reactive cyberaggression (β = 0.25, *p* < 0.001) for the middle school sample. It did not moderate the relationship between self-control and reactive cyberaggression (β = 0.03, *p* = 0.43) for the high school sample. Interestingly, school climate did not moderate the relationship between self-control and instrumental cyberaggression (β = 0.06, *p* = 0.09), but it did moderate the relationship between self-control and reactive cyberaggression (β = 0.10, *p* = 0.01) for the college sample. See Table 3, Table 4 and Table 5 for all model specifications and Table 1, Figure 2, Figure 3 and Figure 4 for the moderating effect.

## 6. Discussion

In this study, we investigated the relationship between spirituality, self-control, and cyberaggression among a group of Chinese middle school, high school, and college students. This is the first study that has ever confirmed the positive relationship between spirituality with self-control, which further negatively correlated with the level of cyberaggression. 

All of our hypotheses have been partially validated by the current data. The mediation model held true across the middle school, high school, and college samples for reactive cyberaggression, and for instrumental cyberaggression, it held true only for the college sample. Therefore, the first and the second hypotheses have been validated. One way to explain the mediation is that spirituality could provide a source of psychological energy [8] which could potentially buffer the self-depletion process and help individuals to access more self-control resources. This would ultimately help individuals better manage their cyberaggression, which is consistent with previous research [23,24,25]. The fact that the mediation did not hold true for instrumental cyberaggression among all samples as reactive cyberaggression did was probably due to the fact that Chinese culture is one that has a low level of tolerance regarding aggression toward others [43], which means people would not engage in aggression unless it is intolerable. When comparing to instrumental cyberaggression, reactive cyberaggression occurs when individuals react when getting hurt. In other words, it is more likely to occur when cyberaggression is intolerable instead of them engaging in cyberaggression actively. With this, it is reasonable to believe these adolescents would consider engaging in less instrumental cyberaggression than they would reactive cyberaggression. Another possibility involves a relatively small middle and high school student sample compared to the college student sample, which potentially leads to nonsignificant results.

School climate has been confirmed to be a protective factor for middle school students’ problematic behaviors [9,10], and it is reasonable to believe that school climate functions as a protective factor for these students against cyberaggression. In terms of the third hypothesis, when the school climate is low, the higher one’s level of spirituality is, the less likely the individual will engage in cyberaggression. It further validated that people who have spirituality usually engage in more moral and prosocial behaviors [8]. 

What is intriguing is the reversed relationship when the school climate is at a higher level. That is, the higher one’s spirituality is, the more likely that one would engage in cyberaggression. On the one hand, for these students, the positive school climate might not have been transferred into their cyber environment, hence reducing its protective effect on cyberaggression. Another possible explanation is that individuals who have spirituality tend to hold high moral standards [7], and these individuals could hold a strong justice world belief [44]. When individuals hold a strong belief in a just world and think that they are treated with justice, they tend to engage in less aggressive behaviors [45], but when they are treated with injustice, they tend to engage in more aggression [46]. If these students hold a strong belief in a just world while encountering injustice, they may feel a higher level of need to counteract injustice when they have a higher level of spirituality, hence engaging in cyberaggression.

What is more, the third hypothesis did not hold true for the high school sample. One possible reason for the result is that high school students in China focus more on how to prepare for their college entrance exam in order to be admitted to a prestigious university. Along with this shared understanding, their sense of belonging to their school, their relationship with their teachers and peers, and the support they receive from their school all decrease to some extent [47,48,49]. The consequence is that the positive impact of school climate becomes less relevant for these high schoolers. On the contrary, middle school students are going through their early adolescent years, during which a sense of belonging is critical [33], and school climate is an essential factor in many areas of their development [32]. In addition, for the lower grade college students, they are at the end of their adolescent years, and a sense of belonging would necessarily resurface after the anxiety ceased along with the college entrance exam being in the past. College students in their senior years, they start to shoulder more and more societal responsibilities in preparation for entering society. One of the important aspects is to have a harmonious relationship with people around them so that they will have more potential resources to use after graduation. We believe these are the main reasons why the third hypothesis held true for the middle school and college student sample but not for the high school one. 

For the fourth hypothesis, when at a lower level of school climate, the lower spirituality is, the lower one’s self-control is, and vice versa. This effect becomes more salient as the level of school climate grows. This is consistent with previous research findings that religiosity could motivate self-control [6]. Since spirituality has a similar function as religiosity does [30], spirituality could motivate self-control for these students as well. Seemingly, a better school climate could enhance the effect of spirituality, such that a better school climate could help one cultivate their spirituality. 

Our last hypothesis was partially validated, such that when the school climate is at a lower level, if individuals’ self-control is low, then they would be more likely to engage in cyberaggression. However, when the school climate is at a higher level, individuals’ cyberaggression is not affected by the level of self-control as much. In other words, self-control does not matter as much when the school climate is high as when the school climate is low. This finding also speaks to the importance of school climate and its protective effect. More importantly, it helps magnify the effect of self-control, which is a very effective factor in managing unethical and aggressive behaviors [4,50].

### Implications and Limitations

The potential implications of our study are as follows. First, it is imperative that school counselors and therapists work with students from the lens of spirituality when it comes to helping these students manage the negative impact of cyberaggression. To compensate for the fact that many interventions that focus on improving self-control are limited in a laboratory setting and have not been transferred to the real world [5], practitioners could attend to help students cultivate their level of spirituality. This is a relevant area as these students are at a stage that is critical to develop and consolidating their outlook on the world, life, and values. Second, schools and colleges need to devote more energy to promoting a more beneficial school climate in which students feel a sense of belonging, have a positive relationship with their peers and teachers and can develop their sense of autonomy [40]. In so doing, the positive impact of spirituality and self-control could be amplified since individuals will be more prone to develop their spirituality and their self-control is higher in such an environment. In terms of implications for future research, experimental studies are often aiming to tackle the direct relationships between stimuli and certain psychological phenomena and, therefore, could be utilized to further test the causal relationship between spirituality, self-control, and cyberaggression directly.

Our study also has some limitations. We have strived to compare group differences between middle school, high school, and college students; however, the sheer number of each sample is far from enabling us to draw any firm conclusions. Future research needs to gather a larger sample in order to replicate our study and test the hypotheses. Our study is of cross-sectional nature, hence reducing the ability to draw a causal conclusion. It is better that future research consider collecting longitudinal data to further validate the relations between these variables. Another limitation is that we measured cyberaggression from the perpetrator’s perspective, leaving out the victims’ viewpoint. The psychological mechanism of completing the action of cyberaggression is complicated. The action is not always initiated by perpetrators but sometimes by those who were once victims [51]. Therefore, we recommend that future researchers measure cyberaggression from both the perpetrator’s and the victims’ perspectives.

## 7. Conclusions

The mediating effect of self-control was significant for the college sample on both types of cyberaggression and marginally significant for the high school and middle school sample on reactive cyberaggression. The moderating effect varied across the three samples. School climate moderated the first half of the mediation model for all three samples, the second half for middle school and college student samples on reactive cyberaggression, the direct path for middle school samples on reactive cyberaggression, and the college student sample on both types of cyberaggression. Spirituality has varying degrees of association with cyberaggression through the mediating role of self-control and the moderating role of school climate.

## Figures and Tables

**Figure 1 ijerph-20-02973-f001:**
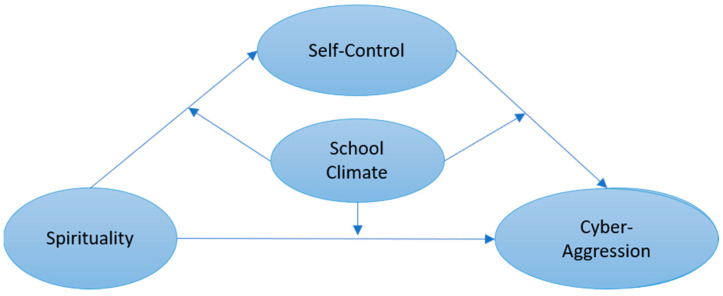
Relationships between Variables.

**Figure 2 ijerph-20-02973-f002:**
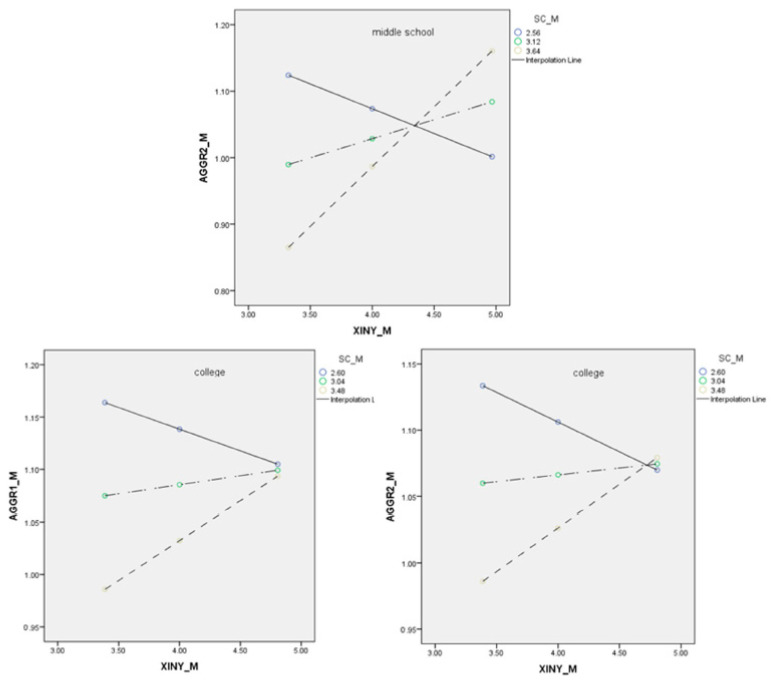
Direct path with school climate being the moderator. Note: XINY = spirituality, AGGR1 = instrumental cyberaggression, AGGR2 = reactive cyberaggression, SC = school climate.

**Figure 3 ijerph-20-02973-f003:**
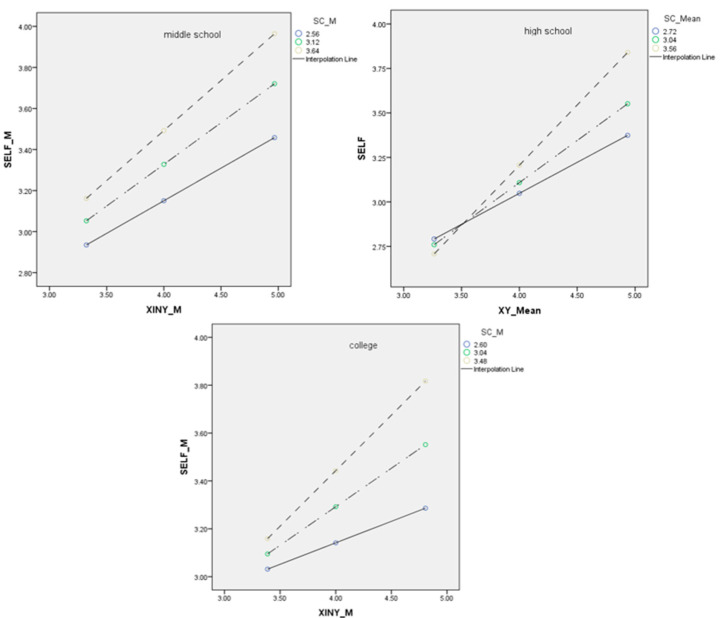
First half path with school climate being the moderator. Note: XINY = spirituality, SELF = self-control, SC = school climate.

**Figure 4 ijerph-20-02973-f004:**
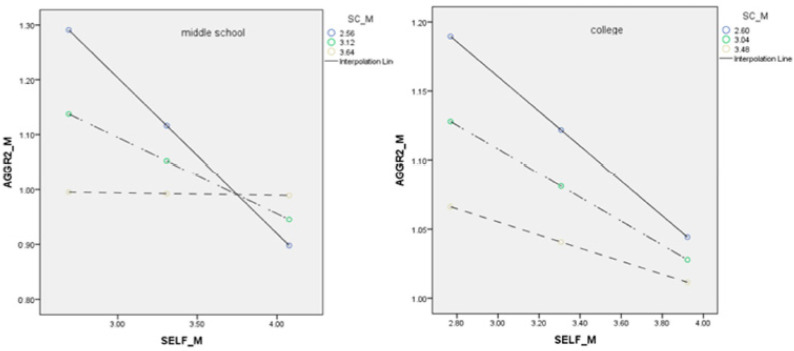
Second half path with school climate being the moderator. Note: SELF = self-control, AGGR2 = reactive cyberaggression, SC = school climate.

**Table 1 ijerph-20-02973-t001:** Descriptive information of the three samples.

	Number and Percentage of Middle School Sample N (%)	Number and Percentage of High School Sample N (%)	Number and Percentage of College Sample N (%)
Sex			
Male	218 (47.8%)	194 (40.8%)	379 (33.9%)
Female	238 (52.2%)	281 (59.2%)	738 (66.1%)
Year			
1st year/Freshman	247 (54.2%)	242 (50.9%)	425 (38%)
2nd year/Sophomore	209 (45.8%)	233 (49.1%)	519 (46.5%)
Junior			124 (11.1%)
Senior			45 (4%)
Other			4 (0.4%)
Number of Siblings			
Single Child	230 (50.4%)	210 (44.2%)	256 (22.9%)
One sibling	197 (43.2%)	214 (45.1%)	597 (53.4%)
Two or more siblings	29 (6.4%)	51 (10.7%)	264 (23.6%)
Parental Monthly Income			
1000 CNY or below	10 (2.2%)	13 (2.7%)	130 (11.6%)
1000 to 5000 CNY	100 (21.9%)	131 (27.6%)	685 (61.3%)
5000 to 10,000 CNY	159 (34.9%)	177 (37.3%)	210 (18.8%)
10,000 CNY or above	187 (41%)	154 (32.4%)	92 (8.2%)
Parental Education Level			
Elementary school or below	7 (1.5%)	12 (2.5%)	193 (17.3%)
Middle school or below	21 (4.6%)	72 (15.2%)	429 (38.4%)
High school or below	99 (21.7%)	171 (36%)	334 (29.9%)
Bachelor’s degree or below	258 (56.6%)	178 (37.5%)	154 (13.8%)
Master’s degree or below	51 (11.2%)	29 (6.2%)	2 (0.2%)
Doctoral degree or below	20 (4.4%)	13 (2.7%)	5 (0.4%)

**Table 2 ijerph-20-02973-t002:** Correlations between variables.

	Spirituality	Self-Control	School Climate	Instrumental Agg	Reactive Agg
Spirituality	--				
Self-Control	0.51	--			
School Climate	0.62	0.45	--		
Instrumental Agg	−0.11	−0.22	−0.20	--	
Reactive Agg	−0.12	−0.22	−0.20	−0.93	--

Note: Instrumental Agg = Instrumental Aggression subscale of the Cyberaggression Behaviors Scale, Reactive Agg = Reactive Aggression subscale of the Cyberaggression Behaviors Scale. All correlations reported here are significant at *p* < 0.001.

**Table 3 ijerph-20-02973-t003:** Indices of Mediation and Moderation models of the Middle School Sample.

	Mediation Model		Moderation of the Direct Path		Moderation of the First Half		Moderation of the Second Half	
	β	*p*	β	*p*	β	*p*	β	*p*
Middle School	Reactive Agg		Reactive Agg		Self-Control		Reactive Agg	
Gender	0.14	=0.003 **	0.06	=0.86	−0.01	=0.83	0.04	=0.14
Age	−0.01	=0.72	−0.03	=0.52	−0.03	=0.36	0.001	=0.95
Grade	0.10	=0.020 **	0.03	=0.95	−0.03	=0.28	0.03	=0.25
Number of Siblings	0.03	=0.67	0.03	=0.88	−0.01	=0.81	−0.006	=0.82
Parental monthly income	−0.001	=0.99	0.01	=0.95	0.04	=0.28	−0.02	=0.05
Parental educational level	0.07	=0.28	−0.01	=0.98	−0.04	=0.24	0.02	=0.47
Spirituality	−0.10	=0.55	−0.68	<0.001 ***	−0.08	=0.58	0.05	=0.08
Self-Control	−0.40	=0.001 ***	0.15	<0.001 ***			−0.94	<0.001 ***
School Climate			−1.0	<0.001 ***	−0.31	=0.11	−0.93	<0.001 ***
Spirituality X School Climate			0.24	<0.001 ***	0.16	=0.001 ***		
Self-Control X School Climate							0.25	<0.0010 ***
R^2^	0.22		0.25		0.33		0.20	

Note. Reactive Agg = Reactive Aggression subscale of the Cyberaggression Behaviors Scale. *** *p* ≤ 0.001, ** *p* ≤ 0.01.

**Table 4 ijerph-20-02973-t004:** Indices of Mediation and Moderation models of the High School Sample.

	Mediation Model		Moderation of the Direct Path		Moderation of the First Half		Moderation of the Second Half	
	β	*p*	β	*p*	β	*p*	β	*p*
Middle School	Reactive Agg		Reactive Agg		Self-Control		Reactive Agg	
Gender	0.15	<0.001 ***	0.09	=0.004 **	−0.05	=0.005 **	0.03	<0.001 ***
Age	−0.03	=0.45	−0.02	=0.40	0.02	=0.55	−0.01	=0.58
Grade	−0.02	=0.72	−0.02	=0.84	0.03	=0.020	−0.01	=0.63
Number of Siblings	−0.03	=0.52	−0.004	=0.80	0.001	=0.96	−0.01	=0.37
Parental monthly income	0.05	=0.16	0.02	=0.21	−0.01	=0.65	0.01	=0.71
Parental educational level	−0.03	=0.70	−0.01	=0.72	0.03	=0.12	−0.004	=0.86
Spirituality	0.23	<0.001 ***	0.05	=0.58	−0.72	<0.001 ***	0.16	<0.001 ***
Self-Control	−0.54	<0.001 ***	−0.16	<0.001 ***			−0.25	=0.049 *
School Climate			−0.28	=0.02 **	−1.38	<0.001 ***	−0.24	=0.09
Spirituality X School Climate			0.04	=0.19	0.39	<0.001 ***		
Self-Control X School Climate							0.03	=0.43
R^2^	0.29		0.12		0.39		0.12	

Note: Reactive Agg = Reactive Aggression subscale of the Cyberaggression Behaviors Scale. *** *p* ≤ 0.001, ** *p* ≤ 0.01, * *p* ≤ 0.05.

**Table 5 ijerph-20-02973-t005:** Indices of Mediation and Moderation models of the College Sample.

	Mediation Model		Moderation of the Direct Path		Moderation of the First Half		Moderation of the Second Half	
	β	*p*	β	*p*	β	*p*	β	*p*
Middle School	Reactive Agg		Reactive Agg		Self-Control		Reactive Agg	
Gender	0.18	<0.001 ***	0.15	<0.001 ***	−0.03	=0.03 *	0.10	<0.001 ***
Age	0.10	=0.25	0.09	=0.26	0.02	=0.43	0.01	=0.53
Grade	−0.04	=0.44	−0.03	=0.51	−0.01	=0.52	−0.01	=0.60
Number of Siblings	−0.02	=0.53	−0.02	=0.57	0.02	=0.23	0.004	=0.74
Parental monthly income	−0.05	=0.12	−0.03	=0.39	0.01	=0.39	−0.004	=0.74
Parental educational level	0.04	=0.34	0.02	=0.68	−0.01	=0.66	0.02	=0.16
Spirituality	−0.10	=0.19	−0.39	<0.001 ***	−0.66	<0.001 ***	−0.01	=0.64
Self-Control	−0.26	=0.02 *	−0.10	<0.001 ***			−0.27	=0.02 *
School Climate			−0.65	<0.001 ***	−0.95	<0.001 ***	−0.32	=0.01 **
Spirituality X School Climate			0.13	<0.001 ***	0.32	<0.001 ***		
Self-Control X School Climate							0.06	=0.09
R^2^	0.13		0.08		0.31		0.06	
College	Reactive Agg		Reactive Agg		Reactive Agg			
Gender	0.17	<0.001 ***	0.15	<0.001 ***	0.09		=0.002 **	
Age	0.13	=0.04	0.12	=0.04	0.02		=0.26	
Grade	−0.05	=0.28	−0.04	=0.34	−0.02		=0.42	
Number of Siblings	−0.04	=0.28	−0.03	=0.30	0.003		=0.83	
Parental monthly income	0.06	=0.04 *	0.04	=0.16	−0.004		=0.77	
Parental educational level	0.01	=0.80	−0.01	=0.80	0.02		=0.25	
Spirituality	0.07	=0.29	−0.37	<0.001 ***	−0.02		=0.45	
Self-Control	−0.31	=0.002 **	−0.10	<0.001 ***	−0.36		=0.002 **	
School Climate			−0.59	<0.001 ***	−0.39		=0.002 **	
Spirituality X School Climate			0.13	<0.001 ***				
Self-Control X School Climate					0.10		=0.01 **	
R^2^	0.16		0.07		0.06			

Note: Instrumental Agg = Instrumental Aggression subscale of the Cyberaggression Behaviors Scale, Reactive Agg = Reactive Aggression subscale of the Cyberaggression Behaviors Scale. *** *p* ≤ 0.001, ** *p* ≤ 0.01, * *p* ≤ 0.05.

## Data Availability

The datasets generated during and analyzed during the current study are not publicly available due to them containing information that could compromise research participant privacy/consent but are available from the corresponding author upon reasonable request.

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
