# Peer review of "Spirituality and Cyberaggression: Mediating and Moderating Effect of Self-Control and School Climate"

_ijerph, 2023, doi:10.3390/ijerph20042973_

Round 1

Reviewer 1 Report

It is necessary to improve English language.

The references could be updated (less than 50% of refereces are from the last 5 years)

Author Response

Thanks to the reviewer’s comments! We have asked a native speaker, who holds a Masters degree in Psychology and works at a college in the USA, to proofread our paper.

In terms of the references, we appreciate the reviewer’s careful reading. We managed to provide a thorough literature review and balance the relatively old versus new research in our reference list. Therefore, we included some old studies on well studies concepts, such as cyberaggression and school climate. In addition, what we found is that for the topic of spirituality, the research studies are relatively old, and we hope that we can add new research studies to the literature pool. We will also be mindful of the reminder in future writing. Thank you again!

Reviewer 2 Report

This paper provides an interesting perspective on a problem that is important to be able to successfully address. Some awkwardness in writing and gaps in logic and depth of descriptions limits its value but could be improved with editing. Specific concerns are listed below.

11.       Abstract, Methods, line 10: change to “We examined 456…”

22.       Abstract, Results, line 12: change to “mediating effect [of what?] was significant for the college sample…”

33.       Abstract, Results, line 14: change to “The moderating effect varied across the three samples.”

44.       Paragraph 1, line 25: delete “and so on.”

55.       Paragraph 1, line 27: change to “Although cognitive resources such as self-control have been identified as important factors that individuals can utilize…”

66.       Paragraph 1, it would be helpful to give a few examples of cyberaggression.

77.       Paragraph 2, line 32, change to “Spirituality has been found to promote…”

88.       Theoretical background, paragraph 1, line 48: change to “and guide each other.”

99.       Theoretical background, paragraph 2, line 55: change to “cyberbullying (Lerner, 2013) because specific factors,…”

110.   Theoretical background, paragraph 2, line 61, last sentence is confusing.

111.   Theoretical background, paragraph 3, line 65, change to: “Interventions to address cyberaggression often utilize cognitive strategies.”

112.   Theoretical background, Spirituality in Chinese Context, line 99, change to “Prior research has attempted to examine the spirituality of Chinese students.”

113.   Theoretical background, Spirituality in Chinese Context, line 103, change to: “(2021) surveyed over 2,000 college students…”

114.   Theoretical background, Spirituality in Chinese Context, paragraph 2, line 111, change to: “higher levels of spirituality are consistent with higher levels of kindness,…”

115.   Theoretical background, Spirituality in Chinese Context, paragraph 2, line 113, change to: “To avoid confusion, the term, “spirituality,” is used in the current study even though “Xin-Yang” might be the more appropriate…”

116.   Theoretical background, Spirituality in Chinese Context, paragraph 2, line 117, change to “No other published studies have been found that examine the relationship between spirituality and self-control and its role in reducing aggression.”

117.   Theoretical background, Spirituality in Chinese Context, it would be helpful to include literature addressing the potential for changing spirituality to support the value of this study. It would also be helpful to add a little more details about the three dimensions from the Zin-Yang scale.

118.    Theoretical background, School Climate Moderating Effect, paragraph 1, line 127, sentence is confusing. Need to rewrite.

119.    Theoretical background, School Climate Moderating Effect, paragraph 1, line 134, change to: “try to intervene in cases of cyberaggression…”

220.   Theoretical background, School Climate Moderating Effect, paragraph 1, line 135, change “levels” to “aspects.”

221.    Theoretical background, School Climate Moderating Effect, paragraph 2,line 137, rewrite sentence 2. It is confusing.

222.   Theoretical background, School Climate Moderating Effect, paragraph 3, line 142, change to: “Similarly, no published study examined school climate as a moderator for the relationship…”

223.   Figure 1, change “Xin-Yang” to “Spirituality.”

224.   Methods, paragraph 2, line 188, write out RMB first time.

225.   Table 1 data do not need to be repeated in text. Add Notes below explaining RMB abbreviation.

226.   Include psychometrics for each assessment tool.

Author Response

  1. Abstract, Methods, line 10: change to “We examined 456…”

Thank you very much for the comments! We appreciate it and now have corrected the wording in the abstract section.

  1. Abstract, Results, line 12: change to “mediating effect [of what?] was significant for the college sample…”

Thank you very much for the comments! We appreciate it and now have corrected the wording in the abstract section.

  1. Abstract, Results, line 14: change to “The moderating effect varied across the three samples.”

Thank you very much for the comments! We appreciate it and now have corrected the wording in the abstract section.

  1. Paragraph 1, line 25: delete “and so on.”

Thanks for the comments! We have now corrected the wording.

  1. Paragraph 1, line 27: change to “Although cognitive resources such as self-control have been identified as important factors that individuals can utilize…”

Thanks for the comments! We have now corrected the wording.

  1. Paragraph 1, it would be helpful to give a few examples of cyberaggression.

Thanks for the comments, but we have provided examples from line 59 to 65.

  1. Paragraph 2, line 32, change to “Spirituality has been found to promote…”

Thank you so much for the recommendation of wording! We have now changed it.

  1. Theoretical background, paragraph 1, line 48: change to “and guide each other.”

Thank you so much for the recommendation of wording! We have now changed it.

  1. Theoretical background, paragraph 2, line 55: change to “cyberbullying (Lerner, 2013) because specific factors,…”

Thank you so much for recommending the change of sentence structure! We have now changed it.

  1. Theoretical background, paragraph 2, line 61, last sentence is confusing.

Thanks to the reviewer’s comments and we have managed to make it clearer.

  1. Theoretical background, paragraph 3, line 65, change to: “Interventions to address cyberaggression often utilize cognitive strategies.”

Thanks to the reviewer’s comments and we have now changed it accordingly.

  1. Theoretical background, Spirituality in Chinese Context, line 99, change to “Prior research has attempted to examine the spirituality of Chinese students.”

Thanks to the reviewer’s comments and we have now changed it accordingly.

  1. Theoretical background, Spirituality in Chinese Context, line 103, change to: “(2021) surveyed over 2,000 college students…”

Thanks to the reviewer’s comments and we have now changed it accordingly.

  1. Theoretical background, Spirituality in Chinese Context, paragraph 2, line 111, change to: “higher levels of spirituality are consistent with higher levels of kindness,…”

Thanks to the reviewer’s comments and we have now changed it accordingly.

  1. Theoretical background, Spirituality in Chinese Context, paragraph 2, line 113, change to: “To avoid confusion, the term, “spirituality,” is used in the current study even though “Xin-Yang” might be the more appropriate…”

Thank you so much for your thoughtfulness and we have changed it now!

  1. Theoretical background, Spirituality in Chinese Context, paragraph 2, line 117, change to “No other published studies have been found that examine the relationship between spirituality and self-control and its role in reducing aggression.”

Thank you so much for your thoughtfulness and we have changed it now!

  1. Theoretical background, Spirituality in Chinese Context, it would be helpful to include literature addressing the potential for changing spirituality to support the value of this study. It would also be helpful to add a little more details about the three dimensions from the Zin-Yang scale.

Thanks to the reviewer’s comments and we have added the information.

  1. Theoretical background, School Climate Moderating Effect, paragraph 1, line 127, sentence is confusing. Need to rewrite.

Thanks to the reviewer’s comments and we have managed to make it clearer.

  1. Theoretical background, School Climate Moderating Effect, paragraph 1, line 134, change to: “try to intervene in cases of cyberaggression…”

Thanks to the reviewer’s comments and we have now changed it accordingly.

  1. Theoretical background, School Climate Moderating Effect, paragraph 1, line 135, change “levels” to “aspects.”

Thanks to the reviewer’s comments and we have now changed it accordingly.

  1. Theoretical background, School Climate Moderating Effect, paragraph 2,line 137, rewrite sentence 2. It is confusing.

Thanks to the reviewer’s comments and we have reworded the sentence to make it clearer.

  1. Theoretical background, School Climate Moderating Effect, paragraph 3, line 142, change to: “Similarly, no published study examined school climate as a moderator for the relationship…”

Thanks to the reviewer’s comments and we have now changed it accordingly.

  1. Figure 1, change “Xin-Yang” to “Spirituality.”

Thanks to the reviewer’s comments and we have now changed it accordingly.

  1. Methods, paragraph 2, line 188, write out RMB first time.

Thanks to the reviewer’s comments and we have now changed it to CNY accordingly.

  1. Table 1 data do not need to be repeated in text. Add Notes below explaining RMB abbreviation.

Thanks to the reviewer’s comments. Since we changed RMB to CNY, which is publicly accepted term for Chinese yuan, we did not include the explanation in the note.

  1. Include psychometrics for each assessment tool.

Thanks to the reviewer’s comments. We have included the information accordingly. Thank you again!

Reviewer 3 Report

Overall, I thought the article was pertinent and presented new information. I also thought the content was well thought out and addressed an area not often discussed.

The only areas I felt that needed improvement was the beginning, sample section, and discussion. 

First, I recommend sectioning out the beginning parts with the following titles, "introduction", "literature review", and "theoretical background"....before leading into methods. There needs to be enough information up front to connect to the results and discussion at the end. 

Second, there also needs to be more discussion on the sample of individuals. Are Chinese students being sampled? It is not clear....middle school and high school students from where? This is critical to discuss so the audience can have a better understanding on the cultural component associated with the sample. 

Third, I recommend to better line out hypotheses 1 and 2 in the discussion so that it's clear that they are being addressed and aren't overlooked. 

Finally, I recommend making sure the paper as a whole is a comprehensive and each section flows well from one to the next. You should be able to connect the introduction to the conclusion, the lit review to the discussion and the methods with the results. There shouldn't be any areas that are misleading or difficult to follow. 

Author Response

Overall, I thought the article was pertinent and presented new information. I also thought the content was well thought out and addressed an area not often discussed.

Thank you for the reviewer’s comments! We appreciate it.

The only areas I felt that needed improvement was the beginning, sample section, and discussion. 

First, I recommend sectioning out the beginning parts with the following titles, "introduction", "literature review", and "theoretical background"....before leading into methods. There needs to be enough information up front to connect to the results and discussion at the end. 

Thanks to the reviewer’s comments. We think it is necessary to add the section headings as suggested. We did so as well.

Second, there also needs to be more discussion on the sample of individuals. Are Chinese students being sampled? It is not clear....middle school and high school students from where? This is critical to discuss so the audience can have a better understanding on the cultural component associated with the sample. 

Thank you so much for the question! We have now added the necessary information.

Third, I recommend to better line out hypotheses 1 and 2 in the discussion so that it's clear that they are being addressed and aren't overlooked. 

We appreciate the reminder from the reviewer and we have added the information now.

Finally, I recommend making sure the paper as a whole is a comprehensive and each section flows well from one to the next. You should be able to connect the introduction to the conclusion, the lit review to the discussion and the methods with the results. There shouldn't be any areas that are misleading or difficult to follow. 

Thank you for the reviewer’s reminder, and we think it is such an important reminder! We have made sure that the paper flows well in terms of the logic and the structure of the study.